# Association of the Transmembrane Serine Protease-2 (TMPRSS2) Polymorphisms with COVID-19

**DOI:** 10.3390/v14091976

**Published:** 2022-09-07

**Authors:** Rosalinda Posadas-Sánchez, José Manuel Fragoso, Fausto Sánchez-Muñoz, Gustavo Rojas-Velasco, Julian Ramírez-Bello, Alberto López-Reyes, Laura E. Martínez-Gómez, Carlos Sierra-Fernández, Tatiana Rodríguez-Reyna, Nora Elemi Regino-Zamarripa, Gustavo Ramírez-Martínez, Joaquín Zuñiga-Ramos, Gilberto Vargas-Alarcón

**Affiliations:** 1Department of Endocrinology, Instituto Nacional de Cardiología Ignacio Chávez, Mexico City 14080, Mexico; 2Department of Molecular Biology, Instituto Nacional de Cardiología Ignacio Chávez, Juan Badiano No. 1, Sección XVI, Tlalpan, Mexico City 14080, Mexico; 3Departament of Immunology, Instituto Nacional de Cardiología Ignacio Chávez, Mexico City 14080, Mexico; 4Intensive Care Unit, Instituto Nacional de Cardiología Ignacio Chávez, Mexico City 14080, Mexico; 5Laboratorio de Gerociencias, Instituto Nacional de Rehabilitación “Luis Guillermo Ibarra Ibarra”, Mexico City 14080, Mexico; 6Dirección de Enseñanza, Instituto Nacional de Cardiología Ignacio Chávez, Mexico City 14080, Mexico; 7Department or Immunology and Rheumatology, Instituto Nacional de Ciencias Medicas y Nutrición Salvador Zubirán, Mexico City 14080, Mexico; 8Tecnologico de Monterrey, Monterrey 64849, Mexico; 9Immunology and Genetics Laboratory, Instituto Nacional de Enfermedades Respiratorias Ismael Cosío Villegas, Mexico City 14080, Mexico

**Keywords:** COVID-19 disease, genetic susceptibility, polymorphisms, SARS-CoV-2 infection, transmembrane serine protease-2 (TMPRSS2)

## Abstract

SARS-CoV-2 uses the ACE2 receptor and the cellular protease TMPRSS2 for entry into target cells. The present study aimed to establish if the TMPRSS2 polymorphisms are associated with COVID-19 disease. The study included 609 patients with COVID-19 confirmed by RT-PCR test and 291 individuals negative for the SARS-CoV-2 infection confirmed by RT-PCR test and without antibodies anti-SARS-CoV-2. Four TMPRSS2 polymorphisms (rs12329760, rs2298659, rs456298, and rs462574) were determined using the 5′exonuclease TaqMan assays. Under different inheritance models, the rs2298659 (*p*_codominant2_ = 0.018, *p*_recessive_ = 0.006, *p*_additive_ = 0.019), rs456298 (*p*_codominant1_ = 0.014, *p*_codominant2_ = 0.004; *p*_dominant_ = 0.009, *p*_recessive_ = 0.004, *p*_additive_ = 0.0009), and rs462574 (*p*_codominant1_ = 0.017, *p*_codominant2_ = 0.004, *p*_dominant_ = 0.041, *p*_recessive_ = 0.002, *p*_additive_ = 0.003) polymorphisms were associated with high risk of developing COVID-19. Two risks (*ATGC* and *GAAC*) and two protectives (*GAGC* and *GAGT*) haplotypes were detected. High levels of lactic acid dehydrogenase (LDH) were observed in patients with the rs462574*AA* and rs456298*TT* genotypes (*p* = 0.005 and *p* = 0.020, respectively), whereas, high heart rate was present in patients with the rs462574*AA* genotype (*p* = 0.028). Our data suggest that the rs2298659, rs456298, and rs462574 polymorphisms independently and as haplotypes are associated with the risk of COVID-19. The rs456298 and rs462574 genotypes are related to high levels of LDH and heart rate.

## 1. Introduction

Coronavirus disease 2019 (COVID-19) is produced by the severe acute respiratory syndrome coronavirus 2 (SARS-CoV-2). Until 28 August 2022, have been confirmed 596,873.121 cases globally and 6459.684 deaths [1].

SARS-CoV-2 uses the angiotensin-converting enzyme-2 (ACE2) receptor and transmembrane serine protease-2 (TMPRSS2) to enter the host cells [2]. Other proteases, such as neutrophil elastase (ELANE), furin, cathepsins, and probably TMPRSS11A [3,4,5] could be involved in this phenomenon.

TMPRSS2 is a member of transmembrane protease serine, a family of proteins with conserved serine protease domains located on the cell membrane. TMPRSS2 is an essential enzyme that cleaves the hemagglutinin of many influenza virus subtypes and the coronavirus protein S [6,7]. TMPRSS2 deficiency protects mice against H1N1 and H7N9 influenza A virus infections [6,8]. TMPRSS2 can cleave the S protein and thus facilitate the entry of SARS-CoV-2 into the cell [2]. Recently has been reported that cell lines expressing TMPRSS2 are highly susceptible to SARS-CoV, MERS-CoV, and SARS-CoV-2 [9]. The gene encoding TMPRSS2 is located on chromosome 21 and is highly polymorphic. The prevalence and mortality of the COVID-19 pandemic show marked geographic variability, suggesting that genetic differences in populations, and the presence of various types of viruses, play an essential role in this variability [10,11,12,13,14]. Recently an in-depth genetic analysis of chromosome 21 using genome-wide association study data established that five polymorphisms within TMPRSS2 and near *MXI* gene were associated with a reduced risk of developing severe COVID-19 [15]. In 2020, our research group performed bioinformatics analysis and reported some possible genes and polymorphisms candidates for study in patients with COVID-19 [16]. In that study, we compared the frequencies of these polymorphisms in several populations, including frequencies of Mexican individuals from Los Angeles, CA, USA. For the present study, we select four polymorphisms (rs12329760, rs2298659, rs456298, and rs462574) with high frequency (more than 15%) in Mexican Americans and with possible functional effects. Thus, the present study aimed to evaluate the association of the TMPRSS2 polymorphisms with COVID-19.

## 2. Materials and Methods

### 2.1. Subjects

The study included 609 COVID-19 patients [58% (*n* = 353) male and 42% (*n* = 256) women, mean age of 50.3 ± 14.56 years] with severe acute respiratory syndrome coronavirus 2 (SARS-CoV-2) infection confirmed by RT-PCR test in at least one biological sample. The patients were recruited from the Instituto Nacional de Cardiología Ignacio Chávez (139 patients), Hospital Juárez de México (108 patients), Instituto Nacional de Rehabilitación Luis Guilleromo Ibarra (162 patients), and Instituto Nacional de Enfermedades Respiratorias Ismael Cosio Villegas (200 patients) during April 2020 to February 2021. The diagnosis of COVID-19 was made considering the presence of fever, diarrhea, fatigue, chills, loss of taste, dry cough, nasal congestion, odor, sore throat, headache, musculoskeletal pain, skin rashes, dizziness, conjunctivitis, heart rate, and oxygen saturation. As a control group, we included 291 individuals negatives for the SARS-CoV-2 infection confirmed by the RT-PCR test [60.1% (*n* = 175) men and 39.0% (*n* = 116) women, mean age of 33.1 ± 7.6 years]. They were personnel (medical residents, nurses, and laboratory personnel) that were in the intensive care unit of the Instituto Nacional de Cardiologia Ignacio Chávez. All these individuals were negative for antibodies anti-SARS-CoV-2 (Elecsys^®^ Anti-SARS-CoV-2, Roche Diagnostics International Ltd. CH-6343, Rotkreuz, Switzerland).

All participants or their relatives signed the institutional consent letter. The study was conducted following the Declaration of Helsinki and approved by the Ethics and Research Committees of the Instituto Nacional de Cardiología Ignacio Chávez (protocol 20–1202, approved 8 January 2021).

### 2.2. Sample Handling

Competent trained personnel handled and processed the patients’ blood samples, which were later transported to the laboratory for processing. Samples were centrifuged, and the serum was separated in a class II biological safety cabinet. Personnel handling the samples used personal protective equipment, including disposable gloves, a lab coat, and a surgical mask. Samples were collected and processed in a laboratory that adheres to the guidelines established in the Official Mexican Standards NOR-007-SSA3-2011, NOM-087-SEMARNAT-SSA1-2002, NOM-010-SSA2-2010, NOM-006-SSA2-2013, and NMX-EC-15189 IMNC-2015.

### 2.3. Biochemical Markers

In COVID-19 patients, creatinine, ferritin, lactic acid dehydrogenase (LDH), C-reactive protein (CRP), total bilirubin, aspartate aminotransferase (AST), alanine aminotransferase (ALT), hemoglobin, and the number of platelets were determined by standard techniques.

### 2.4. Genetic Analysis

Genomic DNA of COVID-19 patients and controls was isolated from 5 mL of peripheral blood (containing EDTA) using a QIAamp DNA Blood Mini kit (QIAGEN, Hilden, Germany). DNA integrity was verified in 1% agarose gels stained with ethyl bromide. Then, DNA quantification was performed using automated spectrophotometry equipment (NanoDrop, ND-1000 spectrophotometer), and aliquots of 10 ng/µL concentration were prepared. Based on the previous results and the functional prediction analysis [16], we selected for the study four TMPRSS2 polymorphisms (rs12329760, rs2298659, rs456298, and rs462574) that were determined using 5′-exonuclease TaqMan genotyping assays, on an ABI Prism 7900HT Fast Real-Time PCR system (Applied Biosystems, Foster City, CA, USA).

### 2.5. Statistical Analysis

Data are expressed as frequencies, median (interquartile range), or mean ± standard deviation. The categorical variables were analyzed using the chi-squared test, whereas the continuous variable comparisons were made by either Student’s *t*-test or Mann–Whitney *U* test, as appropriate. The chi-squared test was used to determine Hardy–Weinberg’s equilibrium. The association of the TMPRSS2 polymorphisms with COVID-19 was analyzed by logistic regression analysis under the inheritance model [additive (major allele homozygotes vs. heterozygotes vs. minor allele homozygotes), codominant 1 (major allele homozygotes vs. heterozygotes), codominant 2 (major allele homozygotes vs. minor allele homozygotes), heterozygote (heterozygotes vs. major allele homozygotes + minor allele homozygotes), dominant (major allele homozygotes vs. heterozygotes + minor allele homozygotes) and recessive (major allele homozygotes + heterozygotes vs. minor allele homozygotes)]. Linkage disequilibrium and haplotype analysis were performed with Haploview software (version 4.1, Broad Institute of Massachusetts Institute of Technology and Harvard University, Cambridge, MA, USA). The biochemical markers in the COVID-19 patients were compared in the different genotypes. The data were expressed as means ± SD, and comparisons were performed by ANOVA and least significant difference (LSD) as a post hoc test; *p* values < 0.05 were considered statistically significant. We used the SPSS software v15.0 (SPSS Chicago, IL) for all analyses.

## 3. Results

### 3.1. Characteristics of the COVID-19 Patients

Six hundred-nine COVID-19 patients were included in the study, with a median age of 50.3 ± 14.56 years. The most frequent comorbidities were obesity in 47.7% (*n* = 293), hypertension in 28.3% (*n* = 174), and T2DM in 26.3% (*n* = 162) (Table 1). The levels of biochemical markers are shown also in this table.

On the other hand, the more common symptoms were cough (64.3%), dyspnea (48.7%), headache (41.3%), fatigue (34.7%), and myalgia (34.7%) (Figure 1).

### 3.2. Distribution of TMPRSS2 Polymorphisms in COVID-19 Patients and Healthy Controls

Figure 2 shows the allele and genotype frequencies of the TMPRSS2 polymorphisms in COVID-19 patients and controls. The observed and expected frequencies of the four polymorphisms were in Hardy–Weinberg equilibrium (*p* > 0.05). Even though the distribution of the rs12329760 was similar in both groups, important differences were observed in the allele and genotype distribution of the rs2298659 (*p* = 0.0054 and *p* = 0.0016, respectively), rs456298 (*p* = 0.0001 and *p* = 0.0006, respectively), and rs462574 (*p* = 0.00002 and *p* = 0.00042, respectively) polymorphisms in COVID-19 patients and healthy controls.

### 3.3. Association of TMPRSS2 Polymorphisms with COVID-19

Under different inheritance models, the rs2298659 (OR = 2.61, 95%CI = 1.30–5.23, *p*_codominant2_ = 0.018; OR = 2.49, 95%CI = 1.26−4.90, *p*_recessive_ = 0.006; OR = 1.38, 95%CI = 1.05–1.80, *p*_additive_ = 0.019), rs456298 (OR = 1.51,95%CI = 1.09–2.09, *p_c_*_odominant1_ = 0.014; OR = 2.27, 95%CI = 1.40–3.69, *p*_codominant2_ = 0.004; OR = 1.65, 95%CI = 1.13–2.42, *p*_dominant_ = 0.009; OR = 1.82, 95%CI = 1.20–2.74, *p*_recessive_ = 0.004; OR = 1.50, 95%CI = 1.18–1.91, *p*_additive_ = 0.0009), and rs462574 (OR = 1.46, 95%CI = 1.07–1.96, *p*_codominant1_ = 0.017; OR = 2.46, 95%CI = 1.42–4.27, *p*_codominant2_ = 0.004; OR = 1.43, 95%CI = 1.02–2.02, *p*_dominant_ = 0.041; OR = 2.24, 95%CI = 1.33–3.17, *p*_recessive_ = 0.002; OR = 1.46, 95%CI = 1.14–1.87, *p*_additive_ = 0.003) polymorphisms were associated with high risk of developing COVID-19 (Table 2).

### 3.4. Haplotype Analysis

The haplotype analysis showed a strong linkage disequilibrium (D’ > 0.85, and r2 > 0.75) between the rs462574, rs456298, rs2298659, and 12329760 polymorphisms. In addition, this analysis revealed four (*ATGC*, *GAAC*, *GAGC*, and *GAGT*) of eight haplotypes with important differences between patients and healthy controls (Table 3). The *ATGC* (OR = 1.51, 95%CI = 1.12–2.05, *p* = 0.006) and *GAAC* (OR = 1.58, 95%CI = 1.03–2.42, *p* = 0.017) haplotypes were associated with risk of developing COVID-19, whereas the *GAGC* (OR = 0.52, 95%CI = 0.39–0.70, *p* < 0.001) and *GAGT* (OR = 0.34, 95%CI = 0.10–1.09, *p* = 0.029) haplotypes were associated with low risk of developing COVID-19. 

### 3.5. Biochemical Markers in COVID-19 Patients according to Different Genotypes

COVID-19 patients with the rs462574 *AA* genotype presented high levels of LDH (380.2 ± 195.5) compared to those with *AG* (352 ± 181.6) and *GG* (308 ± 142.6) genotypes (*p* = 0.005). Patients with this genotype also had elevated heart rate (*AA*: 94.6 ± 20; *AG*: 92 ± 20.5; *GG*: 88 ± 19.4) (*p* = 0.028), and temperature (*AA*: 37 ± 1.52; *AG*: 36.7 ± 0.74; *GG*: 36.5 ± 0.86) (*p* = 0.037). On the other hand, patients with the rs456298 *TT* genotype presented high concentrations of LDH (377 ± 206.3) when compared to carriers of *AT* (33.6 ± 164.4) and *AA* genotypes (314 ± 142.9) (*p* = 0.020), and elevated temperature (37 ± 1.33) compared to patients with *AT* (36.6 ± 0.79) and *AA* genotypes (36.6 ± 0.80) (*p* = 0.001). Patients with the rs456298 *AA* genotype presented high concentrations of total bilirubin (1.28 ± 2.56) when compared to those patients with *AT* (0.82 ± 1.06) and *TT* (0.79 ± 0.76) genotypes (*p* = 0.032). Finally, patients with the rs2298659 *AA* genotype presented elevated temperature (37 ± 0.92) when compared to patient carriers of *GA* (36.6 ± 0.79) and *GG* (36.7 ± 1.09) genotypes (*p* = 0.038) (Table 4).

## 4. Discussion

The role that TMPRSS2 plays in the entry of SARS-CoV-2 into the cell is well known. A comprehensive comparative genetic analysis of 81,000 human genomes suggests the association of TMPRSS2 polymorphisms with COVID-19 susceptibility [17]. In the present study, we analyzed the distribution of four TMPRSS2 polymorphisms in a Mexican cohort of patients with COVID-19. Three polymorphisms (rs2298659, rs456298, and rs462574), and two haplotypes (*ATGC* and *GAAC*) were associated with a high risk of COVID-19. Contrarily, two haplotypes (*GAGC* and *GAGT*) were associated with a low risk of COVID-19. Different distribution of some markers of damage was observed in each genotype of these three polymorphisms.

Variations in genes related to the mechanisms used by SARS-CoV-2 to enter the host cells have an important impact on the variability of the infection observed in different ethnic groups [18]. Several studies that include case-controls, genome-wide association, and some using bioinformatics tools have suggested the participation of some polymorphisms in the susceptibility to COVID-19 [16,19,20,21,22]. In this context, the TMPRSS2 gene has been studied with different results. Recently, Li et al. published a systematic review of *ACE2* and TMPRSS2 polymorphisms associated with COVID-19 [23]. This review included 33 articles with 33,923 patients from 160 regions and 50 countries (principally Caucasians) and identified 12 TMPRSS2 polymorphisms associated with COVID-19. Irham et al. using data from multiple genomes, established that some variants of the TMPRSS2 gene affect the expression of the TMPRSS2 protease in lung tissue, suggesting that these variants could be involved in susceptibility to SARS-CoV-2 infection [24]. Latini et al. analyzed 131 COVID-19 patients by exome sequencing and reported three TMPRSS2 polymorphisms with different distribution in patients than in controls (rs75603675, rs114363287, and rs12329760) [25]. Considering the variability in frequencies of the TMPRSS2 polymorphisms reported associated with COVID-19, we determine only those with possible functional effects and high frequency in a Mexican population from Los Angeles reported previously in a bioinformatics analysis [16]. Only one of the polymorphisms included in our study was analyzed previously in other populations. This polymorphism is the rs12329760, located in the coding region (exon 6) and produces an amino acid change in codon 160 (Val by Met). This polymorphism could be damaging, but it does not seem to have any effect at the post-translational level. The prevalence of this polymorphism varies between 10 and 65%, with higher frequency in Asian populations [16,26]. Wulandari et al. reported an association of rs12329760 with COVID-19 in an Indian population [27]. A similar result was reported by Andolfo et al. in a European genetic ancestry population [15]. On the contrary, a study in a German population did not detect an association of this polymorphism with the risk of infection by SARS-CoV-2 or severity by COVID-19 [28]. This result agrees with our report of no association of this polymorphism with COVID-19 in the Mexican population.

The polymorphisms associated with COVID-19 in our population were rs2298659, rs456298, and rs462574. The rs2298659 polymorphism does not generate an amino acid change (Gly296Gly), but the “in-silico” analysis showed that it could have an impact at the mRNA level, affecting splicing and possibly generating one of the 20 TMPRSS2 isoforms reported so far (https://www.ensembl.org/Homo_sapiens/Transcript/Exons?db=core;g=ENSG00000184012; r = 21:41464300-41531116; t = ENST00000332149) (accessed on 25 June 2022). SRp40 is expressed in all cells, tissues, and organs, including the lung, where airway epithelial cells are found [29,30,31]. The rs456298 and rs462574 are located in the 3′-UTR region and have possible functional effects. The *A* allele of the rs456298 can create a binding site for hsa-miR-450b-5p, whereas the *T* allele disrupts this binding site, increasing the levels of TMPRSS2 mRNA and protein. The expression of miR-450b-5p has been found to be reduced in the lung tissues of bleomycin-treated mice [32]. According to the web-based tool RegulomeDB (https://regulomedb.org/regulame-search/) (accessed on 11 July 2022), this polymorphism affects the state of chromatin with high transcription of the gene in samples from the stomach, mucosa of the rectum, the liver, pancreas, intestine, B cells, lung, natural killer cells, kidney, and heart. In-silico analysis (with the SNP function prediction program) shows that the *G* allele creates binding sites for hsa-miR-127-3p and for hsa-miR-557; thus, the *A* allele disrupts the binding site for has-miR-127-3p and has-miR-557, and it could increase the levels of TMPRSS2 mRNA and protein. These microRNAs could regulate the translation of the TMPRSS2 protease with important effects on the S protein priming of the SARS-CoV-2 and, in consequence, in the process of infection by this virus (Figure 3). Patients with rs462574 *AA* and rs456298 *TT* genotypes presented high concentrations of LDH. In a pooled analysis that included nine studies, Henry et al. reported that elevated LDH concentrations are associated with the severity and mortality of COVID-19 [33]. This result was corroborated in a systematic review of 34 studies [34]. On the other hand, patients with rs462574 *AA* genotype showed a high heart rate, a condition associated with survival chances in patients older than 70 years [35].

The four polymorphisms analyzed were in high linkage disequilibrium, and four haplotypes were associated with COVID-19, two with a high risk (ATGC and GAAC) and two with a low risk (GAGC and GAGT). The two haplotypes associated with low risk included the three alleles with low frequencies in COVID-19 patients (rs462574 G, rs456298 A, and rs2298659 G alleles). The study of haplotypes is important since these haplotypes could be delimiting a region associated with COVID-19.

The association of TMPRSS2 polymorphisms with COVID-19 infection provides not only a reason to evaluate the risk of infection but also a basis for possible prevention or treatment by TMPRSS2 inhibition. Considering the critical role of the TMPRSS2 in the SARS-CoV-2 entry into the cells [2], several inhibitors of this protease have been under study. Nafamostat and camostat are synthetic serine protease inhibitors that can block viral entry to the host cell. In vitro experiments have shown that camostat mesylate and nafamostat mesylate significantly reduce the infection by SARS-CoV-2 [36,37].

A strength of the study was the inclusion of a control group of individuals exposed to the intensive care unit and who were not infected, as demonstrated by the absence of anti-SARS-CoV-2 antibodies. However, some limitations should be considered; (a) the number of controls was smaller than the number of patients, and (b) we only analyzed four polymorphisms of the gene; however, we selected those polymorphisms that, after a bioinformatic analysis, showed possible functional effect and with a frequency of minor allele enough to see statistical differences in our population, and (c) it is not possible to establish if the differences in LDH and heart rate levels are specific to COVID-19 patients or are present in healthy controls, this because these variables were not determined in this later group.

## 5. Conclusions

In conclusion, our data establish that the rs2298659, rs456298, and rs462574 polymorphisms are associated with the risk of COVID-19. We detected two risks (*ATGC* and *GAAC*) and two protective (*GAGC* and *GAGT*) haplotypes. LDH, heart rate, and high temperature were more frequent in patient carriers of specific genotypes of these polymorphisms.

## Figures and Tables

**Figure 1 viruses-14-01976-f001:**
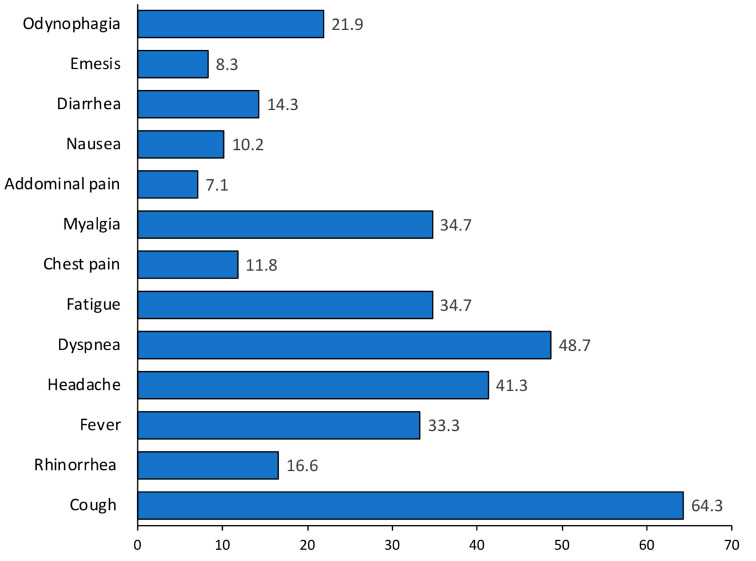
Prevalence of clinical symptoms in COVID-19 patients.

**Figure 2 viruses-14-01976-f002:**
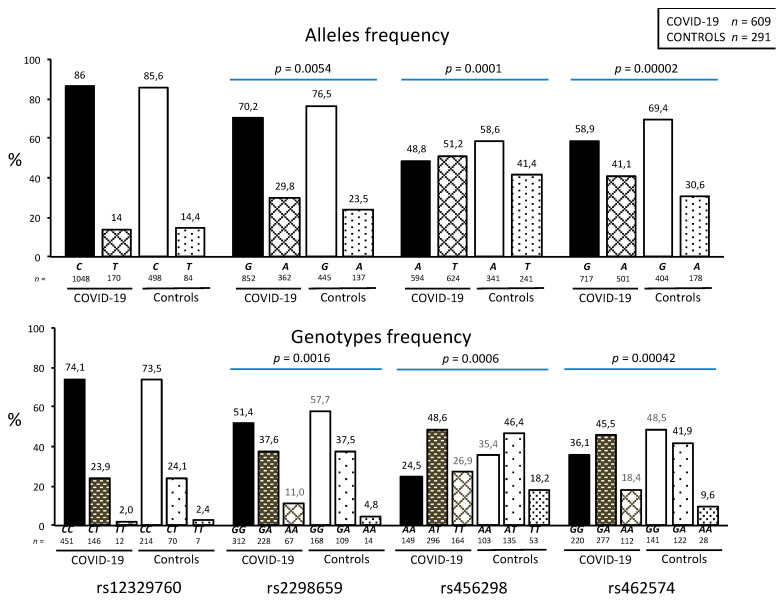
Allele and genotype distribution of TMPRSS2 gene polymorphisms in COVID-19 patients and healthy controls. Data are shown as frequency in percentage. We use Chi-square test to test differences between alleles frequencies and among genotype frequencies. The rs2298659 was determined in 607 COVID-19 patients.

**Figure 3 viruses-14-01976-f003:**
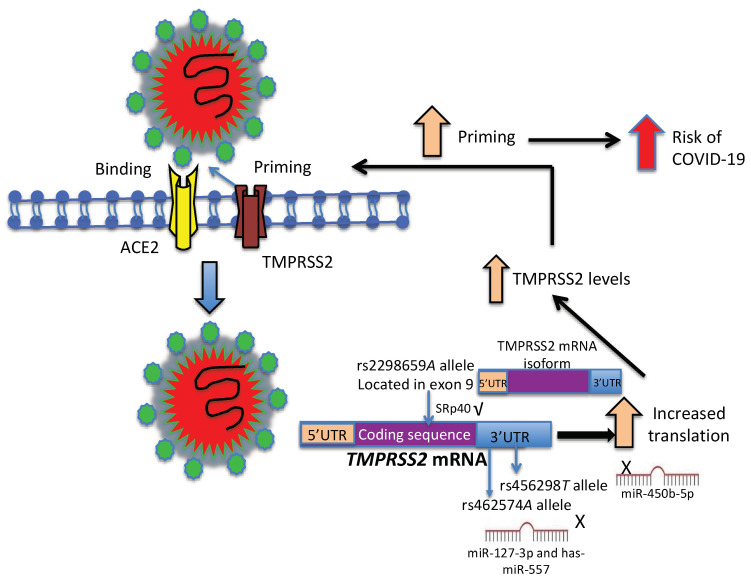
Hypothesis based on in-silico analysis of three SNPs located in the TMPRSS2 gene. TMPRSS2 can cleave the S protein and thus facilitate the entry of SARS-CoV-2 into the cell. According to in-silico analysis, the *A* allele of rs2298659*G/A* could create a binding site for SRp40, a splicing protein involved in the generation of protein isoforms (20 mRNA isoforms have been described). The rs456298*A/T* and rs462574*G/A* polymorphisms (located in 3′ UTR) could create binding sites for miR-450b-5p, miR-1324, and miR-127-3p/miR-557. The rs456298*T* allele of TMPRSS2 could disrupt the binding site for miR-450b-5p, whereas the rs462574*A* allele could disrupt the binding sites for miR-127-3p and miR-557. These alleles could increase the TMPRSS2 protein levels with the consequent increase in priming of the spike protein. This fact could facilitate the entry of SARS-CoV-2 into the cell and increase the risk of COVID-19.

**Table 1 viruses-14-01976-t001:** Clinical data and biochemical parameters of the COVID-19 patients.

Characteristics	COVID-19 Patients (*n* = 609)
Age (years)	50.3 ± 14.56
Sex male *n*, (%)	353, (58)
Temperature (°C)	36.7 ± 0.97
Oxygen saturation (SpO_2_)	86.85 ± 11.28
Heart rate (bpm)	90.94 ± 20.15
Comorbidities	
Obesity *n* (%)	293 (47.7)
TDM2 *n* (%)	162 (26.3)
Hypertension *n* (%)	174 (28.3)
Biochemical markers	
Creatinine (mg/dL)	0.86 [0.68–1.16]
Ferritin (ng/µL)	448 [216–931.7]
LDH (U/dL)	305 [219–406]
Protein C Reactive (mg/dL)	14 [2.89–54.2]
Total bilirubin (mg/dL)	1.0 [0.44–0.87]
ALT (U/dL)	36 [21.1–61.1]
AST (U/dL)	38 [24–81]
Hemoglobin (g/dL)	14 [12.1–15.6]
Platelets (10^9^/L)	257 [199–326]

LDH, Lactic acid dehydrogenase; ALT, Alanine transaminase; AST, aspartate aminotransferase. Data are expressed as mean ± standard deviation, percentage, or median interquartile range.

**Table 2 viruses-14-01976-t002:** Association of TMPRSS2 polymorphisms with COVID-19.

	Genotype Frequency			MAF	Model	OR (95%CI)	*p*
rs12329760	*CC*	*CT*	*TT*	*T*			
Controls (*n* = 291) *n*	0.735 214	0.241 70	0.024 7	0.144	Co-dominant1 Co-dominant2	1.15 (0.77–1.71) 0.86 (0.28–2.69)	0.982 0.749
					Dominant	1.12 (0.77–1.64)	0.549
COVID-19 (*n* = 609)	0.741	0.239	0.020	0.140	Recessive	0.83 (0.27–2.58)	0.751
*n*	451	146	12		Heterozygote	1.16 (0.78–1.71)	0.471
					Additive	1.08 (0.77–1.51)	0.670
rs2298659	*GG*	*GA*	*AA*	*A*			
Controls (*n* = 291) *n*	0.577 168	0.375 109	0.048 14	0.235	Co-dominant1 **Co-dominant2**	1.13 (0.79–1.63) **2.61 (1.30–5.23)**	0.475 **0.018**
					Dominant	1.31 (0.93–1.84)	0.129
COVID-19 (*n* = 607)	0.514	0.376	0.110	0.298	**Recessive**	**2.49 (1.26–4.90)**	**0.006**
*n*	312	228	67		Heterozygote	1.00 (0.70–1.43)	0.978
					**Additive**	**1.38 (1.05–1.80)**	**0.019**
rs456298	*AA*	*AT*	*TT*	*T*			
Controls (*n* = 291) *n*	0.354 103	0.464 135	0.182 53	0.414	**Co-dominant1** **Co-dominant2**	**1.51 (1.09–2.09)** **2.27 (1.40–3.69)**	**0.014** **0.004**
					**Dominant**	**1.65 (1.13–2.42)**	**0.009**
COVID-19 (*n* = 609)	0.245	0.486	0.269	0.512	**Recessive**	**1.82 (1.20–2.74)**	**0.004**
*n*	149	296	164		Heterozygote	0.99 (0.70–1.39)	0.939
					**Additive**	**1.50 (1.18–1.91)**	**0.0009**
rs462574	*GG*	*GA*	*AA*	*A*			
Controls (*n* = 291) *n*	0.485 141	0.419 122	0.096 28	0.306	**Co-dominant1** **Co-dominant**	**1.46 (1.07–1.96)** **2.46 (1.42–4.27)**	**0.017** **0.004**
					**Dominant**	**1.43 (1.02–2.02)**	**0.041**
COVID-19 (*n* = 609)	0.361	0.455	0.184	0.411	**Recessive**	**2.24 (1.33–3.17)**	**0.002**
*n*	220	277	112		Heterozygote	0.97 (0.69–1.37)	0.879
					**Additive**	**1.46 (1.14–1.87)**	**0.003**

MAF, minor allele frequency; OR, odds ratio; CI, confidence interval. The *p*-values were calculated by the logistic regression analysis adjusted by age and gender. In bold are shown the significant models. The rs2298659 polymorphism was determined in 607 COVID-19 patients.

**Table 3 viruses-14-01976-t003:** Frequencies of TMPRSS2 haplotypes in the studied groups.

Haplotype	COVID-19 (*n* = 607)	Controls (*n* = 291)	OR	95%CI	*p*
	Hf (*n*)	Hf (*n*)			
*ATGC*	**0.372 (226)**	**0.289 (84)**	**1.51**	**1.12–2.05**	**0.006**
*GAGC*	**0.278 (169)**	**0.430 (125)**	**0.52**	**0.39–0.70**	**<0.001**
*GAAC*	**0.160 (97)**	**0.110 (32)**	**1.58**	**1.03–2.42**	**0.017**
*GTAT*	0.091 (55)	0.096 (28)	0.95	0.59–1.54	0.860
*GAAT*	0.023 (14)	0.017 (5)	1.38	0.49–3.87	0.538
*GTGC*	0.016 (10)	0.014 (4)	1.22	0.38–3.95	0.729
*GAGT*	**0.008 (5)**	**0.024 (7)**	**0.34**	**0.10–1.09**	**0.029**
*AAGC*	0.013 (8)	0.003 (1)	3.95	0.49–31.8	0.081

Hf, Haplotype frequency. The order of the polymorphisms in the haplotypes is according to the positions in the chromosome (rs462574, rs456298, rs2298659, and 12329760). In bold are shown the significant haplotypes.

**Table 4 viruses-14-01976-t004:** Distribution of the damage biochemical markers in the COVID-19 patients according to the genotypes.

rs462574		Genotypes		*p*
	*GG*	*GA*	*AA*	
Creatinine (mg/dL)	1.35 ± 1.76	1.36 ± 2.51	1.54 ± 1.89	0.801
Ferritin (ng/μL)	612 ± 619.4	624 ± 588.4	764 ± 801	0.208
LDH (U/dL)	**308 ± 142.6**	**352 ± 181.6**	**380.2 ± 195.7**	**0.005**
C reactive protein (mg/dL)	49.4 ± 85.4	51.4 ± 83.9	63.2 ± 103.5	0.537
Total bilirubin (mg/dL)	1.06 ± 2.1	0.87 ± 1.21	0.80 ± 0.84	0.413
ALT (U/dL)	44.6 ± 39.2	53.9 ± 77.2	52.7 ± 47.9	0.393
AST (U/dL)	45.6 ± 32.6	51 ± 74.1	53.8 ± 41.3	0.562
Hemoglobin (g/dL)	13.8 ± 2.96	13.7 ± 2.63	13.6 ± 2.74	0.876
Platelets (10^−6^/µL)	269 ± 115	297 ± 136	280 ± 124	0.113
Heart rate (bpm)	**88 ± 19.4**	**92 ± 20.5**	**94.6 ± 20**	**0.028**
Temperature (°C)	**36.5 ± 0.86**	**36.7 ± 0.74**	**37 ± 1.52**	**0.037**
Oxygen saturation (SpO_2_)	87 ± 11.9	87 ± 11.42	87 ± 9.81	0.999
rs456298				
	*AA*	*AT*	*TT*	
Creatinine (mg/dL)	1.46 ± 2.02	1.26 ± 2.30	1.54 ± 2.04	0.517
Ferritin (ng/µL)	632 ± 640.9	623 ± 588.5	696 ± 737	0.560
LDH (U/dL)	**314.9 ± 142.9**	**336 ± 164.4**	**377 ± 206.3**	**0.020**
C reactive protein (mg/dL)	74.1 ± 82.8	51.6 ± 85.5	59.5 ± 96.2	0.132
Total bilirubin (mg/dL)	**1.28 ± 2.56**	**0.82 ± 1.06**	**0.79 ± 0.76**	**0.032**
ALT (U/dL)	44.1 ± 40.8	54.3 ± 76.9	49.7 ± 43.8	0.416
AST (U/dL)	46.9 ± 35.5	51 ± 73.2	50.1 ± 38.8	0.850
Hemoglobin (g/dL)	14 ± 3.00	14 ± 2.65	13.5 ± 2.74	0.274
Platelets (10^−6^/μL)	269.0 ± 123.5	291 ± 132.7	284 ± 119.4	0.345
Heart rate (bpm)	89.6 ± 19.4	90 ± 20.0	94.4 ± 20.1	0.086
Temperature (ºC)	**36.6 ± 0.80**	**36.6 ± 0.79**	**37 ± 1.33**	**0.001**
Oxygen saturation (SpO_2_)	87 ± 12.04	87 ± 11.5	87 ± 10.3	0.999
rs2298659				
	*GG*	*GA*	*AA*	
Creatinine (mg/dL)	1.45 ± 2.14	1.26 ± 2.24	1.54 ± 2.14	0.642
Ferritin (ng/µL)	702 ± 742.6	601 ± 539.2	536 ± 478.9	0.174
LDH (U/dL)	342 ± 171.2	345 ± 185.1	334 ± 152.7	0.930
C reactive protein (mg/dL)	50 ± 82.9	58.9 ± 97.2	50.1 ± 79.3	0.634
Total bilirubin (mg/dL)	0.90 ± 1.29	0.90 ± 1.23	1.04 ± 2.79	0.846
ALT (U/dL)	48.4 ± 43.42	54.9 ± 85.5	45.2 ± 39.4	0.527
AST (U/dL)	49.4 ± 39.3	52.3 ± 81.9	42.6 ± 26.4	0.619
Hemoglobin (g/dL)	13.7 ± 2.83	13.68 ± 2.59	14.1 ± 2.99	0.658
Platelets (10^−6^/µL)	277 ± 125.9	292 ± 121.7	292 ± 147.7	0.467
Heart rate (bpm)	92 ± 20.4	89 ± 19.1	91 ± 21.5	0.314
Temperature (°C)	**36.7 ± 1.09**	**36.6 ± 0.79**	**37 ± 0.92**	**0.038**
Oxygen saturation (SpO_2_)	87 ± 10.82	88 ± 11.8	84 ± 11.8	0.086

LDH = lactic acid dehydrogenase, ALT = aminotransferase alanine, AST = aminotransferase aspartate. Data are expressed as mean ± SD. In bold are shown significant differences.

## Data Availability

The data presented in this study are available upon request from the corresponding author.

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
