# Peer review of "Association of the Transmembrane Serine Protease-2 (TMPRSS2) Polymorphisms with COVID-19"

_viruses, 2022, doi:10.3390/v14091976_

Round 1
Reviewer 1 Report
The authors compared four polymorphisms of TMPRSS2, a protease important for SARS-CoV-2 infection into airway epithelial cells, between control and patient groups and found significant differences in three polymorphisms.
The rs2298659 is a polymorphism that is located in a coding region but does not cause amino acid mutations, and in silico analysis suggested that it is involved in SRp40 protein binding.
On the other hand, the polymorphisms rs456298, and rs462574 in the 3' UTR were suggested to be involved in the formation of several microRNA recognition sequences.
From these information, the authors speculated that the three polymorphisms contribute to SARS-CoV-2 infection by regulating the expression level of TMPRSS2. Results on the involvement of these polymorphisms in COVID-19 varied by cohort, suggesting differences in ethnic backgrounds and other factors. The biological analysis of the effect of the polymorphism on the expression level of TMPRSS2 was not performed and is only descriptive.
As important as the present results are in terms of the association between COVID-19 and TMPRSS2 polymorphisms in Mexicans, they would be more valuable with the addition of some supporting information below.
1. Are there differences in expression and function among ethnic groups for some microRNAs that have been suggested to be associated with the TMPRSS2 polymorphism?
2. Are these microRNAs and SRp40 expressed and functioned in airway epithelial cells?
3. Are differences in LDH and Heart rate limited to COVID-19 patients? Or are there differences in healthy controls as a potential factor?
4. The association of TMPRSS2 polymorphisms with COVID-19 infection provides not only a reason to evaluate the risk of infection, but also a basis for possible prevention or treatment by TMPRSS2 inhibition. Prospects for treatment and prevention by using TMPRSS2 inhibitor such as camostat, nafamostat etc. should be discussed.
Author Response
1.- Are there differences in expression and function among ethnic groups for some microRNAs that have been suggested to be associated with the TMPRSS2 polymorphism?
Answer: Unfortunately, there are not information about the expression or function of the microRNAs associated with the TMPRSS2 among populations.
2.- Are these microRNAs and SRp40 expressed and functioned in airway epithelial cells?
Answer: Some information about the expression of microRNAs, and SRp40 has been reported. This information has been included and the following phrases have been added in the discussion section.
“SRp40 is ubiquitously expressed in all cells, tissues, and organs, including the lung, where airway epithelial cells are found [29–31]”
- Diamond, R.; Du, K.; Lee, V.; Mohn, K.; Haber, B.; Tewari, D.; Taub, R. Novel Delayed-Early and Highly Insulin-Induced Growth Response Genes. Identification of HRS, a Potential Regulator of Alternative Pre-MRNA Splicing - PubMed. J Biol Chem 1993, 268, 15185–15192.
- Du, K.; Taub, R. Alternative Splicing and Structure of the Human and Mouse SFRS5/HRS/SRp40 Genes. Gene 1997, 204, 243–249, doi:10.1016/S0378-1119(97)00552-0.
- Kim, H.R.; Lee, G.O.; Choi, K.H.; Kim, D.K.; Ryu, J.S.; Hwang, K.E.; Na, K.J.; Choi, C.; Kuh, J.H.; Chung, M.J.; et al. SRSF5: A Novel Marker for Small-Cell Lung Cancer and Pleural Metastatic Cancer. Lung Cancer 2016, 99, 57–65, doi:10.1016/J.LUNGCAN.2016.05.018.
“The expression of miR-450b-5p has been found to be reduced in lung tissues of bleomycin-treated mice [32].”
- Lin, S.; Zhang, R.; Xu, L.; Ma, R.; Xu, L.; Zhu, L.; Hu, J.; An, X. LncRNA Hoxaas3 Promotes Lung Fibroblast Activation and Fibrosis by Targeting MiR-450b-5p to Regulate Runx1. Cell Death Dis. 2020, 11, doi:10.1038/S41419-020-02889-W.
3.- Are differences in LDH and Heart rate limited to COVID-19 patients? Or are there differences in healthy controls as a potential factor?
Answer: Unfortunately, the LDH levels and heart rate only was determined in patients with COVID-19. This is considered a limitation of the study. The phrase “it is not possible to establish if the differences in LDH and heart rate levels are specific to COVID-19 patients or are present in healthy controls, this because these variables were not determined in this later group.” has been included in the discussion section.
4.- The association of TMPRSS2 polymorphisms with COVID-19 infection provides not only a reason to evaluate the risk of infection, but also a basis for possible prevention or treatment by TMPRSS2 inhibition. Prospects for treatment and prevention by using TMPRSS2 inhibitor such as camostat, nafamostat etc. should be discussed.
Answer: As comment the reviewer, the use of TMPRSS2 inhibitors as treatment of COVID-19 is under study. To discusses this issue, the phrase “The association of TMPRSS2 polymorphisms with COVID-19 infection provides not only a reason to evaluate the risk of infection but also a basis for possible prevention or treatment by TMPRSS2 inhibition. Considering the critical role of the TMPRSS2 in the SARS-CoV-2 entry into the cells [2], several inhibitors of this protease have been under study. Nafamostat and camostat are synthetic serine protease inhibitors that can block viral entry to the host cell. In vitro experiments have shown that camostat mesylate and nafamostat mesylate significantly reduce the infection by SARS-CoV-2 [36,37].” has been added in the discussion section.
- McKee, D.L.; Sternberg, A.; Stange, U.; Laufer, S.; Naujokat, C. Candidate Drugs against SARS-CoV-2 and COVID-19. Pharmacol. Res. 2020, 157, doi:10.1016/J.PHRS.2020.104859.
- Wang, M.; Cao, R.; Zhang, L.; Yang, X.; Liu, J.; Xu, M.; Shi, Z.; Hu, Z.; Zhong, W.; Xiao, G. Remdesivir and Chloroquine Effectively Inhibit the Recently Emerged Novel Coronavirus (2019-NCoV) in Vitro. Cell Res. 2020, 30, 269–271, doi:10.1038/S41422-020-0282-0.
Reviewer 2 Report
In the manuscript entitled “Association of the transmembrane serine protease-2 (TMPRSS2) polymorphisms with COVID-19.” The authors aimed to establish if the TMPRSS2 polymorphisms are associated with COVID-19 disease. It is interesting study, however there are still some scientific criticisms need to be addressed by the authors to strengthen the manuscript.
1. The authors did not conduct experiments (e.g., Luciferase assay) on whether the genotype and haplotype of three SNPs regulate mRNA expression of TMPTSS2 gene. So, I think it would be better to delete the hypothesis of Figure 3.
2. The authors need to further explain why they chose four SNPs among several SNPs of TMPRSS2 gene.
3. The authors need to provide LD and HWE for these four SNPs.
4. It is better to write not only the percentage of samples but also the number of samples in material and methods section, Figure 2, Table 2, and Table 3.
5. What are the criteria that authors decided COVID-19 samples are severe cases?
6. Authors need to explain what is different between co-dominant1 and 2.
7. Authors need to provide more details briefly about the genetic analysis section of how isolated genomic DNA…etc.
8. English editing and proofreading through the entire manuscript need to be done.
For example,
Line 34, rs456298, y rs462574
Line 42, poly-morphisms
Line 50-51, the sentence needs to be rewrite.
Line 72-74, the sentence needs to be rewrite.
Line 131, co-dominat2
Line 144, Biochemical, and clinical parameters
Line 178, In table 3 shows the haplotype frequencies
Line 247, in silico
Line 249, rs456298, and rs462574
Line 252, hsa-miR-135a y b
Etc.
Author Response
1.- The authors did not conduct experiments (e.g., Luciferase assay) on whether the genotype and haplotype of three SNPs regulate mRNA expression of TMPTSS2 gene. So, I think it would be better to delete the hypothesis of Figure 3.
Answer: We agree with the reviewer that the functional effect of the studied polymorphisms is supported only by bioinformatics analysis. However, in this context, we considered that is a hypothesis that can explain the participation of these SNPs in the SARS-CoV-2 infection. In the legend of the figure, we clarify that this hypothesis is based on predictions on in silico analysis. However, if the reviewer disagrees with this, we can delete the figure.
The legend of the figure has been modified: Hypothesis based on in-silico analysis of three SNPs located in the TMPRSS2 gene. TMPRSS2 can cleave the S protein and thus facilitate the entry of SARS-CoV-2 into the cell. According to in-silico analysis, the A allele of rs2298659G/A could create a binding site for SRp40, a splicing protein involved in the generation of protein isoforms (20 mRNA isoforms have been described). The rs456298A/T and rs462574G/A polymorphisms (located in 3’ UTR) could create binding sites for miR-450b-5p, miR-1324, and miR-127-3p/miR-557. The rs456298T allele of TMPRSS2 could disrupt the binding site for miR-450b-5p, whereas the rs462574A allele could disrupt the binding sites for miR-127-3p and miR-557. These alleles could increase the TMPRSS2 protein levels with the consequent increase of priming of the spike protein. This fact could facilitate the entry of SARS-CoV-2 into the cell and increase the risk of COVID-19.
2.- The authors need to further explain why they chose four SNPs among several SNPs of TMPRSS2 gene.
Answer: In order to clarify this point, the phrase “In 2020, our research group performed a bioinformatics analysis and reported some possible genes and polymorphisms candidates for study in patients with COVID-19 [16]. In that study, we compared the frequencies of these polymorphisms in several populations, including frequencies of Mexican individuals from Los Angeles, CA, USA. For the present study, we select four polymorphisms (rs12329760, rs2298659, rs456298, and rs462574) with high frequency (more than 15%) in Mexican Americans and with possible functional effects.” has been included in the introduction section.
3.- The authors need to provide LD and HWE for these four SNPs.
Answer: The D´ and r2 for the haplotypes has been included. The phrase “The haplotype analysis showed a strong linkage disequilibrium (D’ > 0.85, and r2 > 0.75) between the rs462574, rs456298, rs2298659, and 12329760 polymorphisms. In addition, this analysis revealed four (ATGC, GAAC, GAGC, and GAGT) of eight haplotypes with important differences between patients and healthy controls (Table 3).” has been included in the section 3.4. Haplotype analysis.
Also, the phrase “Observed and expected frequencies of the four polymorphisms were in Hardy-Weinberg equilibrium (p > 0.05).” has been added in the section 3.2. Distribution of TMPRSS2 polymorphisms in COVID-19 patients and healthy controls.
4.- It is better to write not only the percentage of samples but also the number of samples in material and methods section, Figure 2, Table 2, and Table 3.
Answer: We included the number of samples in material and methods, Figure 2, Table 2, and Table 3.
5.- What are the criteria that authors decided COVID-19 samples are severe cases?
Answer: We did not evaluate severity of the disease. The severe acute respiratory syndrome coronavirus 2 (SARS‐CoV‐2) refers to the virus. The diagnosis of COVID-19 was made considering the presence of fever, diarrhea, fatigue, chills, loss of taste, dry cough, nasal congestion, odor, sore throat, headache, musculoskeletal pain, skin rashes, dizziness, conjunctivitis, heart rate, and oxygen saturation.
6.- Authors need to explain what is different between co-dominant1 and 2.
Answer: The genotypes compared in each model have been included in the statistical analysis section. The phrase “The association of the TMPRSS2 polymorphisms with COVID-19 was analyzed by logistic regression analysis under the inheritance model [additive (major allele homozygotes vs. heterozygotes vs. minor allele homozygotes), codominant 1 (major allele homozygotes vs. heterozygotes), codominant 2 (major allele homozygotes vs. minor allele homozygotes), heterozygote (heterozygotes vs. major allele homozygotes + minor allele homozygotes), dominant (major allele homozygotes vs. heterozygotes+minor allele homozygotes) and recessive (major allele homozygotes+heterozygotes vs. minor allele homozygotes)].” has been added.
7.- Authors need to provide more details briefly about the genetic analysis section of how isolated genomic DNA…etc.
Answer: More information about the DNA extraction has been added. The phrase “Genomic DNA of COVID-19 patients and controls was isolated from 5 ml. of peripheral blood (containing EDTA) using a QIAamp DNA Blood Mini kit (QIAGEN, Hilden, Germany). DNA integrity was verified in 1% agarose gels stained with ethyl bromide. Then, DNA quantification was performed using automated spectrophotometry equipment (NanoDrop, ND-1000 spectrophotometer), and aliquots of 10 ng/ul. concentration were prepared.” has been added in material and methods.
8.- English editing and proofreading through the entire manuscript need to be done.
Answer: The English has been revised and corrected.
Round 2
Reviewer 2 Report
This revised manuscript was satisfactorily modified for my comments. Thus, I am recommending acceptance for publication in the Viruses.